# Optimization of Plant Growth Regulators for In Vitro Mass Propagation of a Disease-Free 'Shine Muscat' Grapevine Cultivar

Si-Hong Kim [1,2,†], Mewuleddeg Zebro [3,†], Dong-Cheol Jang [1,4], Jeong-Eun Sim [3], Han-Kyeol Park [1], Kyeong-Yeon Kim [1], Hyung-Min Bae [5], Shimeles Tilahun [6,7,*] and Sung-Min Park [1,4,*]

[1] Interdisciplinary Program in Smart Agriculture, Kangwon National University, Chuncheon 24341, Republic of Korea; tlghdek@kangwon.ac.kr (S.-H.K.); jdc@kangwon.ac.kr (D.-C.J.); fallphk@kangwon.ac.kr (H.-K.P.); rlaruddus09@naver.com (K.-Y.K.)
[2] Smart Farm Research Center, KIST Gangneung, Institute of National Products, 679 Saimdang-ro, Gangneung 25451, Republic of Korea
[3] Department of Plant Science, Gangneung-Wonju National University, Gangneung 25457, Republic of Korea; zmewuledeg@gmail.com (M.Z.); jamma2659@naver.com (J.-E.S.)
[4] Department of Horticulture, Kangwon National University, Chuncheon 24341, Republic of Korea
[5] Novagreen Business Centre, Kangwon National University, Chunchen 24341, Republic of Korea; novagreen2109@gmail.com
[6] Agriculture and Life Science Research Institute, Kangwon National University, Chuncheon 24341, Republic of Korea
[7] Department of Horticulture and Plant Sciences, Jimma University, Jimma 378, Ethiopia
* Correspondence: shimeles@kangwon.ac.kr (S.T.); parksm@kangwon.ac.kr (S.-M.P.); Tel.: +82-332506424 (S.M.P.)
† These authors contributed equally to this work.

**Abstract:** This study addresses the propagation challenges faced by 'Shine Muscat', a newly introduced premium grapevine cultivar in South Korea, where multiple viral infections pose considerable economic loss. The primary objective was to establish a robust in vitro propagation method for producing disease-free grapes and to identify effective plant growth regulators to facilitate large-scale mass cultivation. After experimentation, 2.0 µM 6-benzyladenine (BA) exhibited superior shoot formation in the Murashige and Skoog medium compared with kinetin and thidiazuron. Conversely, α-naphthaleneacetic acid (NAA) hindered shoot growth and induced callus formation, while indole-3-butyric acid (IBA) and indole-3-acetic acid (IAA) demonstrated favorable root formation, with IBA showing better results overall. Furthermore, inter simple sequence repeat analysis confirmed the genetic stability of in vitro-cultivated seedlings using 2.0 µM BA and 1.0 µM IBA, validating the suitability of the developed propagation method for generating disease-free 'Shine Muscat' grapes. These findings offer promising prospects for commercial grape cultivation, ensuring a consistent supply of healthy grapes in the market.

**Keywords:** auxin; cytokinin; ISSR marker; in vitro; 'Shine Muscat'

## 1. Introduction

Grapevines, belonging to *Vitis* species, are economically important fruits with a wide range of varieties [1]. They are known for their high content of beneficial compounds such as resveratrol and anthocyanins, which possess anticancer, antioxidant, and anti-inflammatory properties [2,3]. However, grapevine viruses, fruit quality deterioration, and seedling necrosis present significant challenges to grape cultivation, particularly in Central and Southeast Asi [4]. To address these challenges, breeders are developing hybrids of European grapes (*V. vinifera*) and American grapes (*V. labrusca*) to enhance disease and pest resistance in vines.

'Shine Muscat', a diploid-bred grape cultivated in Japan, closely resembles European species and the yellow–green Alexandrian-type grape [5]. In Korea, the cultivation of Shine Muscat, an interspecific diploid hybrid of *V. labruscana* Bailry and *V. vinifera* L., has been rapidly expanding due to increasing consumer preference [6,7]. This cultivar is known for

its high carbohydrate content and can be enjoyed by peeling the skin, similar to European grapes. 'Shine Muscat' has gained popularity in Korea and shares similar characteristics with cultivars developed in Japan [7,8].

While the cultivation areas of commonly grown grape varieties such as 'Campbell Early', 'Geobong', and 'MBA' are decreasing, there is a notable increase in the cultivation areas of new high-quality varieties like 'Shine Muscat' [6]. However, a study by Kim et al. [9] reported a high prevalence of grapevine viruses (91.0%) in the primary grape-producing regions of Korea, regardless of the specific region or cultivar.

To ensure the competitiveness of the grapevine industry, establishing a virus-free and disease-free grapevine cultivation system is crucial. In vitro culture plays a vital role in this regard, and it requires the identification of appropriate medium composition [10]. The in vitro culture method consists of two stages: shoot growth and rooting. Previous studies have highlighted the influence of different types and concentrations of plant growth regulators (PGRs) on shoot growth, rooting efficiency, and genotype variations in grapevines [11,12]. Therefore, the efficiency of in vitro propagation can vary depending on the genetic characteristics of the species, necessitating the evaluation and selection of optimal PGR conditions.

In *Vitis* species and other plants, specific cytokinins and auxins play distinct roles in promoting shoot, bud, or root formation during in vitro culture. Cytokinins, such as 6-benzyladenine (BA), kinetin, and zeatin, induce shoot proliferation by promoting the development of multiple shoots from explants, including shoot tips and nodal segments [13,14]. They stimulate cell division and meristem growth, resulting in the formation of new shoots [15]. Additionally, cytokinins help in bud initiation and outgrowth, leading to the development of lateral shoots [16]. On the other hand, auxins like indole-3-butyric acid (IBA), indole-3-acetic acid (IAA), and naphthalene acetic acid (NAA) are crucial for root induction and elongation [17]. They stimulate the formation of adventitious roots from various explants, including shoots and leaf tissues [18]. Moreover, endogenous auxins in *Vitis* species regulate the intricate balance between shoot and root development by influencing various growth processes, such as apical dominance, root formation, and adventitious root initiation [19]. Their precise transport and distribution within the plant, facilitated by PIN proteins, contribute to the spatial arrangement of shoots and roots [20,21]. Understanding the role of endogenous auxins is essential for optimizing growth and development strategies, such as in vitro micropropagation, where exogenous application of auxins can manipulate shoot and root formation for desired outcomes [22]. The balance between cytokinins and auxins is essential in regulating shoot-to-root conversion, as a higher cytokinin-to-auxin ratio promotes shoot development, while a higher auxin-to-cytokinin ratio induces rooting [23,24].

Additionally, in vitro plant propagation methods have been associated with the occurrence of somaclonal mutations [25–27]. Since grape is a perennial crop, genetic variations resulting from in vitro propagation can only be observed at maturity and fruiting stages [28]. Therefore, early detection of genetic variations in in vitro-raised plants is crucial. Molecular markers like ISSR (inter-simple sequence repeats), ISS (interspersed repetitive sequence), RAPD (random amplified polymorphic DNA), and AFLP (amplified fragment length polymorphism) are commonly used to assess genetic variation in plants. ISSR involves amplifying specific regions between microsatellite repeats, providing a cost-effective and easy-to-use method to evaluate genetic diversity [29]. ISS, on the other hand, targets interspersed repetitive elements, enabling researchers to study different genomic regions [30]. RAPD utilizes random primers to amplify DNA fragments, making it a quick and less expensive technique for genetic diversity analysis [31]. AFLP utilizes restriction enzymes and PCR to create DNA fragments, offering high reproducibility and sensitivity [32].

In this study, the genetic stability of 'Shine Muscat' was assessed using inter simple sequence repeat (ISSR) markers. ISSR markers provide a reliable and easily performed method for evaluating genetic stability in grapes as demonstrated in previous studies [33,34]. Hence, the objective of this study was to determine the optimal PGR composition for mass prop-

agation using in vitro culture and to assess the genetic stability of in vitro-raised plants using ISSR markers, aiming to facilitate the large-scale commercial use of 'Shine Muscat'.

## 2. Materials and Methods

### 2.1. Culture Establishment and Media Composition

Fifty healthy 'Shine Muscat' plants, cultivated in a greenhouse for one year, were selected as the source materials for this study. Nodal explants, each carrying a 2 cm axillary bud, were carefully collected from actively growing young shoots, located approximately three to six nodes below the tip. Aseptic techniques were used to establish the in vitro culture. Initially, the nodal segments were washed with running tap water for ten minutes, followed by surface sterilization. This involved immersing the segments in 70% ethanol for 1 min and subsequently treating them with 2% (*w/v*) NaOCl for 10 min. To remove NaOCl, the segments were rinsed five times with sterile distilled water in a fume hood.

To initiate the culture, the sterilized nodal segments were placed in Pyrex rimless glass culture tubes (22 mm × 175 mm) containing $1\times$ MS medium. The medium was supplemented with various concentrations of cytokinins (6-benzyladenine, kinetin, and thidiazuron) ranging from 1.0 to 16.0 μM and compared with the control (MS medium). Additionally, sucrose was added to the medium at a concentration of 30 g L$^{-1}$, and Difco Bacto agar was used to solidify it at a concentration of 8 g L$^{-1}$. After an initial culture period of four weeks, shoot organogenesis was assessed for each treatment, with thirty explants included in each group. Throughout the culture duration, controlled conditions were maintained, including a temperature of 25 ± 1 °C, a light intensity of 36.59 μmol m$^{-2}$s$^{-1}$ photosynthetic photon flux density (PPFD), and a photoperiod of 16 h of daylight.

### 2.2. Shoot Multiplication and Shoot-Related Data Collection

Following the initial four weeks of culture, micro-cuttings obtained from the first-generation in vitro shoots were transferred to another culture tube. The subculture medium composition remained unaltered, except for the omission of kinetin (KIN), and was compared with the control (MS medium). Subsequently, these subcultured micro-cuttings were subjected to the same controlled culture conditions as previously described, aiming to identify the most favorable conditions for multiple shoot proliferation. The evaluation involved recording the number of shoots and nodes per explant, as well as measuring the main shoot length after three weeks of subculture. Each treatment for shoot multiplication was replicated four times, and each replication comprised six explants.

### 2.3. Root Induction, Root-Related Data Collection, and Acclimatization

To investigate the influence of plant growth regulators (PGRs) on root formation, the nodal segments of the shoots were cut to 1 cm, and the shoots were subsequently subcultured in MS medium supplemented with 2.0 μM BA. Micro-cuttings measuring over 0.5 cm in length were then transferred to culture vessels containing different types of auxins or PGR-free media. The experiment involved testing three types of auxins (IAA, IBA, NAA) at concentrations of 0.25, 0.5, 1.0, 2.0, and 4.0 μM to identify the most effective PGR conditions for root induction compared with the control (MS medium). The same culture conditions utilized for shoot proliferation were maintained throughout the experiment. After four weeks of rooting treatments, data on growth parameters were collected following the methodology established by Kim et al. [9]. Each root induction treatment was replicated four times, with six explants included in each replication.

Subsequently, plants that successfully developed roots were transferred to a 72-hole plug tray, utilizing a peat moss-to-perlite ratio of 1:1 (*v/v*), and covered with a plastic container. To ensure successful acclimatization, the humidity level was maintained at 90–95% and gradually reduced over three weeks. During this period, the plants were kept at a constant temperature of 23 ± 1 °C and received a light intensity of 53.66 μmol m$^{-2}$s$^{-1}$ PPFD for 12 h daily. Once the plants were sufficiently hardened off, they were transplanted

into 1 L plastic pots filled with a suitable potting medium and placed in the glasshouse at Kangwon National University, Republic of Korea.

*2.4. Genetic Stability Analysis Using ISSR Markers*

In this study, ten explants derived from the optimal in vitro culture concentration were randomly chosen for evaluating genetic stability utilizing ISSR markers. Total genomic DNA was isolated from the young leaf tissue of each plant following the TaKaRa MiniBEST Plant Genomic DNA Extraction Kit instructions. The concentration of the extracted DNA was diluted to 50 ng $\mu L^{-1}$, as determined with a nanodrop spectrophotometer (MicroDigital, Seongnam, Republic of Korea).

For the ISSR analysis, a preliminary experiment with 15 ISSR primers (UBC primer Set No. 9, University of British Columbia, Canada) was conducted, and subsequently, 10 primers yielding clear bands were selected for further ISSR analysis (Table 1). The PCR reaction solution consisted of 2 µL of genomic DNA, Accupower premix (Bioneer, Daejeon, Republic of Korea), and 10 pmol primers, making a total volume of 20 µL. PCR amplifications were carried out in a Dice® Touch thermal cycler (TaKaRa, Otsu, Japan) with the following cycling parameters: initial denaturation at 94 °C for 5 min, followed by 35 cycles of denaturation at 94 °C for 30 s, annealing at 48, 50, or 55 °C for 45 s, amplification at 72 °C for 2 min, and a final extension step of 7 min at 72 °C. The resulting PCR amplicons were separated using 1.5% agarose gel electrophoresis and visualized using a GD-1000 gel documentation system (Axygen, San Francisco, CA, USA). Specific fragments within the size range of 100 to 2000 base pairs, consistently generated and clearly resolved, were selected for analysis. These fragments were then utilized to identify the presence or absence of ISSR markers in each sample.

**Table 1.** List of ISSR primers used in this study and number of band classes generated.

| ISSR Primer | Annealing Temperature (°C) | Nucleotide Sequence (5′-3′) | No. of Distinct Band Classes | Total Number of Bands Amplified | % Similarity |
|---|---|---|---|---|---|
| UBC 808 | 50 | $(AG)_8C$ | 8 | 80 | 100 |
| UBC 812 | 50 | $(GA)_8C$ | 8 | 80 | 100 |
| UBC 815 | 55 | $(CT)_8G$ | 4 | 40 | 100 |
| UBC 823 | 50 | $(TC)_8C$ | 4 | 40 | 100 |
| UBC 825 | 55 | $(AC)_8T$ | 9 | 90 | 100 |
| UBC 836 | 50 | $(AG)_7ACYA$ | 8 | 80 | 100 |
| UBC 840 | 50 | $(GA)_8Y$ | 8 | 80 | 100 |
| UBC 873 | 50 | $(GACA)_4$ | 13 | 130 | 100 |
| UBC 878 | 55 | $(GGAT)_4$ | 8 | 80 | 100 |
| UBC 881 | 53 | $GGGT(GGGGT)_2G$ | 8 | 80 | 100 |

To assess the genetic relationships among the samples, Jaccard's similarity coefficient was calculated to generate a similarity matrix. A cluster analysis was performed using the unweighted pair group method with arithmetic averages (UPGMA), and a dendrogram was constructed using NTSYS-PC Ver. 2.1 software.

*2.5. Statistical Analysis*

Statistical analyses were conducted using SPSS 25.0, a software program (SPSS Inc., Chicago, IL, USA). Analysis of variance was used to assess the effects of the treatments. When the treatment effects were statistically significant ($p < 0.05$), means were compared using Duncan's multiple range test.

## 3. Results and Discussion

This study addresses the propagation challenges faced by the 'Shine Muscat' grapevine cultivar in South Korea. The primary objectives were to establish a robust in vitro propagation method for producing disease-free grapes and to identify effective plant growth

regulators to facilitate large-scale mass cultivation. An overall summary of this study is shown in Figure 1.

**Figure 1.** Graphic summary of this study.

### 3.1. Assessment of the Impact of Hormones on the Initiation of Culture

The outcomes of the initial responses of the nodal segments to different cytokinins after a four-week incubation period are presented in Table 2. Within two weeks, sprouting buds became apparent in the nodal segments. The regeneration rate of the nodal segments varied based on the type and concentration of cytokinins used in the experiment, ranging from 30.4% to 100% (Table 2). Nodal segments cultured in a cytokinin-free medium did not display any shoot or bud formation. However, when nodal segments were cultivated in media containing cytokinins, callus formation was observed after ten days, and the development of new axillary buds was frequently noted after three weeks. This investigation contradicts the study conducted by Murashige and Skoog [35]. This discrepancy could arise from variations in plant responses to nutrients and growth regulators. Unique physiological and biochemical traits inherent to different plant species might contribute to their distinct reactions to combinations of nutrients and cytokinins. Another potential explanation lies in the intricate nature of plant morphogenesis, governed by a multitude of factors such as signaling pathways, gene expression, and hormonal interactions [36]. Nevertheless, additional research is required to validate the results obtained in the present study. Throughout the four-week duration, the medium supplemented with BA exhibited the highest response in terms of shoot and node formation. The regeneration rate with varying concentrations of BA in the culture medium ranged from 90.4% to 97.8%, and there were no significant differences observed. Notably, an increase in BA concentration led to a corresponding increase in the number of shoots and nodes per nodal segment.

The initial culture efficiency of the medium with TDZ addition ranged from 86.4% to 94.5%, which was slightly lower than that of the BA-supplemented medium, but the difference was not significant. In contrast, the medium treated with KIN displayed a lower survival rate of 37.8% to 67.4% compared with the cytokinin-containing medium, and it showed no significant difference from the non-treated section without cytokinins. Prior studies on various types of vines have also demonstrated that KINs do not yield favorable results compared with other cytokinin hormones [37,38]. Therefore, the application of various concentrations of KIN in this study did not have a positive effect on the regeneration or sprout induction of 'Shine Muscat' compared with BA and TDZ.

**Table 2.** Effect of cytokinin type and concentration of plant growth regulators on culture establishment in the 'Shine Muscat' cultivar.

| Plant Growth Regulators/Concentrations | Bud Induction Rate/Nodal Segment (%) | No. of Shoots/ Nodal segment | No. of Nodes/ Nodal Segment | Length of Main Shoot (cm) |
|---|---|---|---|---|
| Control (MS medium) | 54.3 b | 0.72 d | 2.31 d | 1.31 c |
| BA 1.0 μM | 90.4 a | 1.05 bc | 3.79 abc | 2.07 ab |
| BA 2.0 μM | 97.8 a | 1.21 abc | 4.29 a | 2.37 a |
| BA 4.0 μM | 92.3 a | 1.31 ab | 4.32 a | 2.21 ab |
| BA 8.0 μM | 93.7 a | 1.38 a | 4.43 a | 1.93 ab |
| BA 16.0 μM | 94.3 a | 1.29 ab | 4.57 a | 1.62 bc |
| KIN 1.0 μM | 53.7 b | 0.67 d | 2.27 d | 1.69 bc |
| KIN 2.0 μM | 60.0 b | 0.65 d | 2.84 bcd | 1.81 bc |
| KIN 4.0 μM | 43.7 b | 0.57 d | 2.47 d | 2.24 ab |
| KIN 8.0 μM | 37.5 b | 0.51 d | 2.34 d | 1.91 ab |
| KIN 16.0 μM | 30.4 b | 0.43 d | 1.97 b | 1.57 bc |
| TDZ 1.0 μM | 89.4 a | 0.97 c | 3.64 abc | 1.97 ab |
| TDZ 2.0 μM | 92.3 a | 1.04 bc | 4.21 a | 2.35 a |
| TDZ 4.0 μM | 94.5 a | 1.20 abc | 4.32 a | 2.32 a |
| TDZ 8.0 μM | 92.4 a | 1.09 bc | 4.01 a | 2.26 ab |
| TDZ 16.0 μM | 86.4 a | 0.94 c | 3.81 abc | 2.14 ab |

Note: Means followed by the same letter within each column are not significantly different at $p < 0.05$.

In the present study, both BA and TDZ showed promising results, with BA exhibiting the highest response in terms of shoot and node formation. This finding aligns with the studies conducted on '*Dracocephalum forrestii*' and 'Rhododendron', where cytokinins have been widely used and proven effective in inducing shoot proliferation [14,15]. Cytokinins play a crucial role in promoting cell division, bud formation, and shoot elongation, which are essential for successful micropropagation [39]. On the other hand, KIN was less effective in promoting regeneration and bud formation in 'Shine Muscat'. KINs have been reported to have limited effects on shoot induction and multiplication compared with other cytokinins [40]. The lower survival rate observed with KIN treatments in this study is in line with previous studies that have suggested its limited utility for in vitro propagation [41].

*3.2. Assessment of the Impact of Hormones on the Growth and Multiplication of Shoots*

Additional investigations were undertaken to identify the most effective concentrations of cytokinins, specifically, BA and TDZ, for shoot multiplication, building on the positive outcomes observed during the initial culture phase. The subculture responses of the explants obtained from the initial cultures displayed variations depending on the type and concentration of cytokinins, as outlined in Table 3. Although there were no significant differences between BA and TDZ, media supplemented with BA demonstrated a good response (0.2% to 3.7%). Although the untreated plot induced 0.41 shoots, the medium containing BA and TDZ led to the induction of 1.04 to 1.89 shoots. Notably, the formation of additional buds and nodes exhibited significant differences based on the type and concentration of cytokinin used. Comparatively, media supplemented with cytokinins resulted in a higher number of shoots and nodes compared with cytokinin-free media at all concentrations. This effect could be due to the stimulation of endogenous auxins, as cytokinins induce the synthesis of auxins.

TDZ showed a positive impact on the main shoot development, and there were no adverse effects on shoot and node formation. The subculture responses in media containing TDZ ranged from 83.3% to 89.7% in terms of shoot induction, with the number of shoots, nodes, and length of the main shoot varying from 1.04 to 1.58, 4.32 to 5.27, and 1.48 to 1.58, respectively. Other criteria exhibited an increasing trend as the TDZ concentration reached 8.0 μM, followed by a decline at higher concentrations. It is important to note that high cytokinin concentration with TDZ could lead to shoot fascination and inhibition

of shoot elongation in grapevines, although these aspects were not investigated in this study. Nevertheless, previous studies have reported the positive effects of TDZ on multiple shoot formation in various Vitis species, suggesting its potential usefulness in the in vitro propagation of 'Shine Muscat' [22,23]. Therefore, these findings indicate that TDZ, like BA, can offer valuable benefits for the in vitro propagation of 'Shine Muscat'.

**Table 3.** Effect of cytokinin type and concentration of plant growth regulators on shoot multiplication in the 'Shine Muscat' cultivar.

| Plant Growth Regulators/Concentrations | Bud Induction Rate/Nodal Segment (%) | No. of Shoots/Nodal Segment | No. of Nodes/Nodal Segment | Length of Main Shoot (cm) |
|---|---|---|---|---|
| Control (MS medium) | 42.1 b | 0.41 d | 1.38 c | 0.51 b |
| BA 1.0 μM | 93.2 a | 1.88 abc | 5.24 ab | 1.53 ab |
| BA 2.0 μM | 92.1 a | 2.14 a | 6.51 a | 1.69 a |
| BA 4.0 μM | 84.3 a | 1.89 abc | 5.87 ab | 1.45 ab |
| BA 8.0 μM | 82.9 a | 1.54 bc | 5.54 ab | 1.33 ab |
| BA 16.0 μM | 81.5 a | 1.31 bcd | 5.42 ab | 1.21 ab |
| TDZ 1.0 μM | 89.7 a | 1.04 c | 4.32 b | 1.48 ab |
| TDZ 2.0 μM | 85.4 a | 1.24 bc | 4.47 ab | 1.51 ab |
| TDZ 4.0 μM | 83.3 a | 1.31 bcd | 4.87 ab | 1.55 ab |
| TDZ 8.0 μM | 81.9 a | 1.58 bc | 5.27 ab | 1.58 ab |
| TDZ 16.0 μM | 81.3 a | 1.39 bcd | 5.09 ab | 1.49 ab |

Note: Means followed by the same letter within each column are not significantly different at $p < 0.05$.

Throughout the initial culture period, subculture responses in media containing BA exhibited consistent performance across all concentrations. Moreover, the number of buds and nodes increased at low concentrations, especially at 2.0 μM BA. Among the various hormonal conditions tested, the highest number of shoots and nodes per explant was achieved with media containing 2.0 μM BA, resulting in 2.14 shoots and 6.51 nodes per explant. These values were 1.35- and 1.24-fold higher, respectively, than TDZ at 8.0 μM. The optimal types and concentrations of cytokinins for in vitro propagation can vary significantly among different plant species [24,25]. However, previous studies have emphasized the positive effects of BA on multiple shoot formation in various plants [26,27]. In grapevines, the ideal concentration of BA for shoot multiplication ranged from 2.22 to 11.1 μM. Concentrations above this range inhibited shoot growth and promoted callus formation, reducing the number of shoots and nodes [28]. Another study reported that a BA concentration of 2.5 μM yielded the best results in terms of growth per explant, while concentrations higher than 2.5 μM decreased these values [42]. Importantly, in this study, the initial culture efficiency at 2.0 μM BA was satisfactory, and when subcultured with 2.0 μM BA, the shoots derived from these cultures exhibited uniform growth and healthy green leaves. These findings strongly support the appropriateness of using 2.0 μM BA for sustainable in vitro propagation of 'Shine Muscat'.

This research emphasizes the significance of cytokinins, specifically BA and TDZ, in promoting shoot multiplication in 'Shine Muscat' grapevines. Both BA and TDZ supplementation led to a higher number of shoots and nodes compared with media without cytokinins. These findings align with previous studies that extensively used cytokinins for in vitro shoot multiplication [13,14,16]. Furthermore, this study's findings on 'Shine Muscat' grapevines validate the beneficial effects of TDZ on main shoot development and multiple shoot formation, consistent with earlier research [43,44]. TDZ is well-known for its ability to stimulate shoot proliferation in various woody plants, as demonstrated in previous studies [45,46].

*3.3. Investigation of the Influence of Hormones on the Initiation and Development of Roots*

'Shine Muscat' explants were subcultured using 2.0 μM BA, and the response to root growth was evaluated by treating them with different types and concentrations of auxins. The outcomes demonstrated variations in root formation based on the type and

concentration of auxins used, as presented in Table 4. When plants produced with 2.0 μM BA were cultured in a medium without any plant growth regulator, a lower rate of root formation was observed. Additionally, the lower part of the explants turned black within two weeks or experienced decreased survival rates toward the late stage of culture in the auxin-free medium.

**Table 4.** Effect of auxin type and concentration of plant growth regulators on root induction in the 'Shine Muscat' cultivar.

| Plant Growth Regulators/Concentrations | % Rooting | % Callusing | Root Number | Root Length (cm) | Shoot Length (cm) |
|---|---|---|---|---|---|
| Control (MS medium) | 37.4 b | 0 d | 0.39 d | 0.68 c | 0.57 c |
| IAA 0.25 μM | 69.7 ab | 0 d | 2.18 bcd | 1.37 a | 1.11 abcd |
| IAA 0.50 μM | 70.5 ab | 0 d | 2.49 abcd | 1.24 ab | 1.18 abcd |
| IAA 1.0 μM | 70.8 ab | 0 d | 2.59 abcd | 1.18 abc | 1.23 abcd |
| IAA 2.0 μM | 67.8 ab | 0 d | 2.71 abcd | 1.04 abc | 1.46 abc |
| IAA 4.0 μM | 65.3 ab | 0 d | 2.54 abcd | 0.87 bc | 1.39 abc |
| IBA 0.25 μM | 81.1 a | 0 d | 2.29 bcd | 1.12 abc | 1.53 ab |
| IBA 0.50 μM | 83.7 a | 0 d | 2.79 abcd | 1.24 ab | 1.47 abc |
| IBA 1.0 μM | 86.8 a | 0 d | 3.42 ab | 1.29 ab | 1.63 a |
| IBA 2.0 μM | 70.4 ab | 11.8 cd | 2.57 abcd | 0.91 bc | 1.22 abcd |
| IBA 4.0 μM | 69.7 ab | 24.8 c | 1.84 c | 0.89 bc | 1.09 bcd |
| NAA 0.25 μM | 66.1 ab | 47.5 b | 1.80 c | 0.84 bc | 1.08 bcd |
| NAA 0.50 μM | 68.2 ab | 63.7 ab | 2.61 abcd | 1.04 abc | 0.89 de |
| NAA 1.0 μM | 70.3 ab | 70.5 ab | 2.97 abcd | 1.12 abc | 0.00 f |
| NAA 2.0 μM | 71.8 ab | 73.8 ab | 3.71 a | 0.94 abc | 0.00 f |
| NAA 4.0 μM | 75.7 ab | 80.2 a | 3.34 abc | 0.73 c | 0.00 f |

Note: Means followed by the same letter within each column are not significantly different at $p < 0.05$.

Among the three auxins tested at various concentrations, 1.0 μM IBA displayed the highest rooting percentage (86.8%) and the second-highest number of roots (3.42) without any callus formation (Table 4). At the same concentration level, IAA and NAA produced 2.59 and 2.97 roots with a reduced rooting rate of 16 and 16.5%, respectively, compared with IBA. However, NAA treatments induced callus formation, and higher concentrations of NAA significantly inhibited shoot growth compared with the other auxins. Conversely, IAA did not lead to callus formation or growth inhibition and hyperhydricity at lower concentrations, but the rooting rate and other important criteria were lower than those of IBA.

In this study, the introduction of exogenous auxin led to the initiation of new primordia formation, specifically promoting the development of "xylem pole" vasculature. The establishment of this specialized vasculature played a crucial role in facilitating directional auxin transport for de novo root formation [47]. The observed differences in root formation among the auxins in 'Shine Muscat' are consistent with findings from other studies on woody plants, including Vitis sp. [48,49]. The choice of the appropriate auxin for root induction varies among plant species, and IBA is often preferred due to its more consistent and reliable rooting response [50]. The negative effects of NAA on shoot and root induction align with reports of its toxicity and callus formation issues in certain plant species [51].

Extensive studies have demonstrated that the rooting of explants in woody plants is greatly influenced by the type and concentration of auxins used [29]. Optimal conditions for auxin-induced rooting can vary among plant species and genotypes. While NAA application has shown positive results for root induction in some species, it has also been associated with issues such as toxicity and callus formation, leading to unsuccessful acclimatization of plantlets in other studies [30,31]. Our study observed similar side effects with NAA application in 'Shine Muscat', including callus formation and failure of shoot and root induction, which worsened at higher concentrations. These findings are consistent with previous reports and suggest a potential genetic sensitivity of 'Shine Muscat' to NAA toxicity. Additionally, the IAA application was found to be less effective than IBA for root induction in 'Shine Muscat'. These results could be attributed to the physical properties of

IAA, as it can decompose under the medium and light conditions used, leading to reduced interactions between cytokinins and auxin hormones [32]. While further investigations are required to fully understand these observations, our results suggest that the application of IBA may be more effective for the in vitro propagation of 'Shine Muscat', offering stability, persistence, and protection against toxicity when following our protocol.

This study highlights the significance of using suitable hormones for the successful cultivation of plants during in vitro propagation. Figure 2 illustrates the effects of different plant growth regulators, specifically cytokinins and auxins, on shoot development and root growth. A survival test involving fifty plantlets of 'Shine Muscat' produced with 2.0 µM BA for shoot induction and 1.0 µM IBA during acclimatization (unpublished data) showed that only three plantlets died, indicating no significant survival issues. These findings underscore the effectiveness of utilizing 2.0 µM BA and 1.0 µM IBA for shoot multiplication and root induction in the in vitro propagation of 'Shine Muscat'.

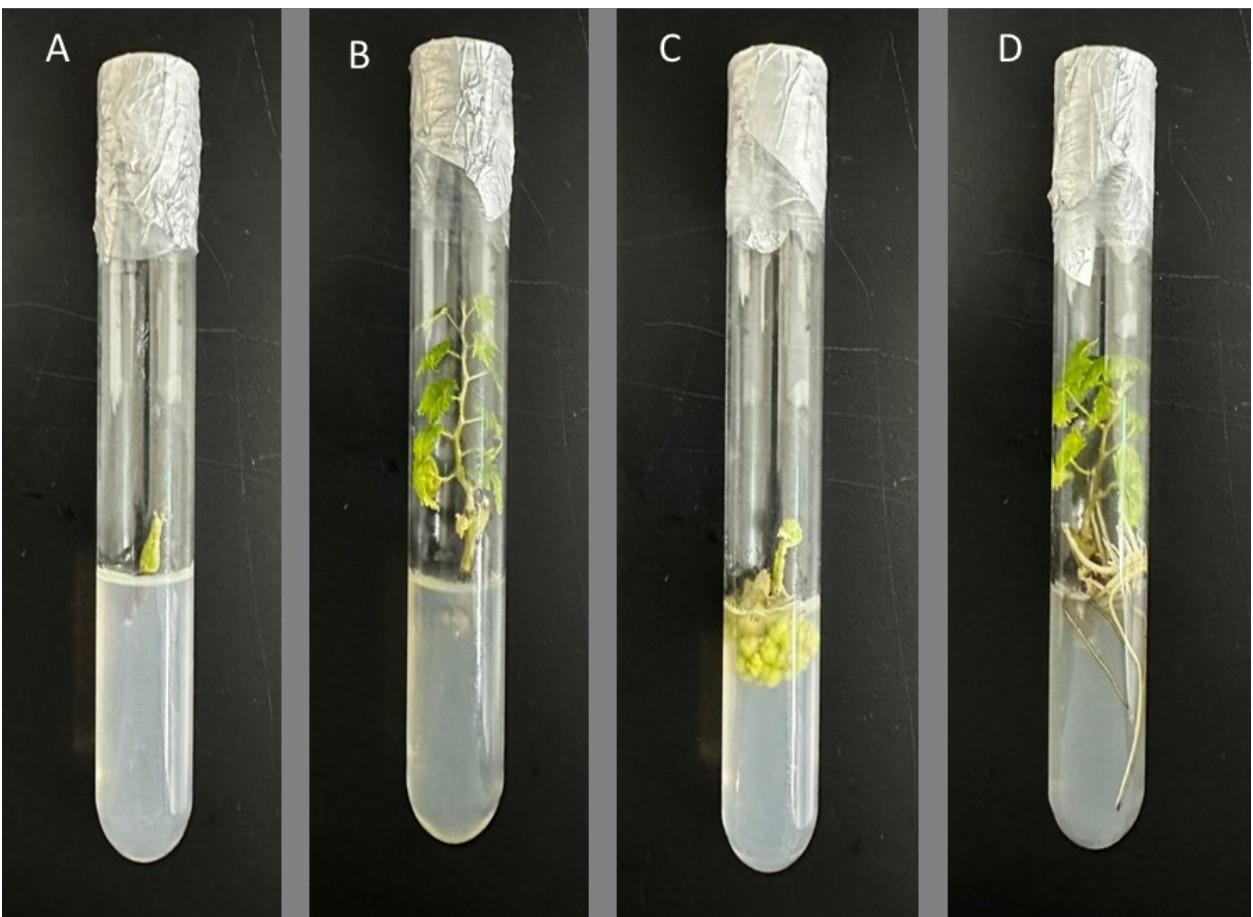

**Figure 2.** Photographs showing the effects of different plant growth regulators, specifically, cytokinins and auxins, on shoot development and root growth: (**A**) KIN at 16.0 µM resulted in the lowest shoot number; (**B**) BA at 2 µM led to the highest shoot number; (**C**) NAA at 4.0 µM resulted in the highest percentage of callus formation; (**D**) IBA at 1.0 µM led to the highest percentage of root formation.

*3.4. Evaluation of Genetic Stability of 'Shine Muscat' Using ISSR Markers*

To assess the genetic stability of ten in vitro-grown plants, ISSR markers were used, and their profiles were compared to the parental plants. Out of the 10 ISSR primers tested, all successfully amplified genomic DNA, resulting in 78 distinct and scorable bands ranging in size from 100 to 2000 bp, with an average of 7.4 bands per primer. The ISSR analysis revealed homogeneity between the regenerated plants and the parental plants, as shown in Figure 3. These results indicate that 'Shine Muscat' maintained its genetic stability throughout in vitro propagation, with no noticeable morphological differences observed.

These findings align with previous studies that have shown genetic uniformity among in vitro-raised plantlets of various plant species [33,34].

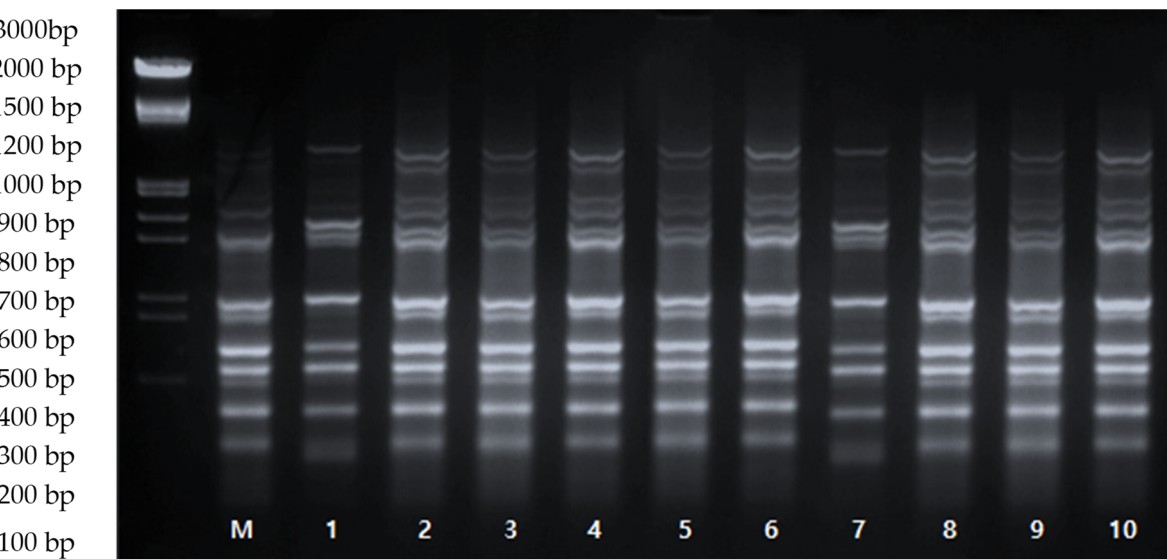

**Figure 3.** ISSR amplification profiles of the 'Shine Muscat' grape cultivar using primer UBC-873. M: mother plant, 1–10: in vitro-raised plants using 2.0 μM BA and 1.0 μM IBA.

The occurrence of variations during in vitro propagation depends significantly on the choice of explants and the regeneration method used [35]. Nodal segments are commonly preferred due to their high regeneration efficiency, and plants derived from adventitious buds within the nodal segment show a low risk of genetic variation [24,36]. However, the excessive use of synthetic plant growth hormones can potentially induce somatic mutations in the obtained tissues [37]. In this study, ISSR markers proved to be valuable tools for assessing the genetic fidelity of the in vitro-regenerated plants. The genetic profiles determined using DNA markers demonstrated that genetic fidelity was well-maintained during somatic embryogenesis, with only rare instances of deviation [38]. Moreover, the analysis of the hormonal conditions using ISSR markers revealed no differences between the in vitro-obtained plants and the parental plant, indicating that the hormone treatment used did not lead to significant genetic deviations. Thus, the results of the ISSR analysis provide robust evidence supporting the genetic stability of 'Shine Muscat' during in vitro propagation. The homogeneity observed between the regenerated plants and the parental plants confirms the effectiveness of the in vitro culture method using nodal segments as explants. This finding is consistent with previous studies reporting genetic stability among in vitro-raised plantlets of various plant species [33,34], which is crucial for preserving desired traits during commercial propagation of elite varieties.

The use of ISSR markers as a tool for evaluating genetic stability is particularly valuable due to their advantages over other molecular marker techniques like RAPD, ISS, or AFLP. ISSR markers do not require prior knowledge of the target DNA sequence and are highly versatile, making them applicable to a wide range of plant species [29,52].

## 4. Conclusions

The successful production of genetically identical plants using in vitro plant propagation holds significant importance. In our study, we discovered that using 2.0 μM BA for shoot induction and 1.0 μM IBA for root induction in 'Shine Muscat' resulted in highly effective plantlet production. This hormone treatment led to a higher number of shoots and roots compared with the hormone-free medium and other types of growth hormone applications. Notably, when using 2.0 μM BA for shoot induction and 1.0 μM IBA for root induction, no instances of somaclonal mutation were observed between the in vitro-grown plants. We utilized ISSR markers to identify somatic mutations, and the results confirmed

the genetic stability of the propagated plants. Consequently, our established protocol for in vitro plant propagation can be reliably used to ensure the essential genetic stability required for the commercial multiplication of the 'Shine Muscat' cultivar.

**Author Contributions:** S.-H.K. and M.Z.: Conducted the experiments and wrote the original manuscript draft. J.-E.S., H.-K.P., K.-Y.K. and H.-M.B.: Helped conduct the experiment and write the original manuscript draft. S.T.: Writing—review and editing. D.-C.J. and S.-M.P.: Conceptualization, supervision, funding acquisition, and writing—review and editing. All authors have read and agreed to the published version of the manuscript.

**Funding:** This study was supported by a 2023 Research Grant from Kangwon National University and financial support from the Ministry of Education (NRF- 2022R1l1A1A01054769).

**Institutional Review Board Statement:** Not applicable.

**Informed Consent Statement:** Not applicable.

**Data Availability Statement:** All data sets are available upon reasonable request from the corresponding author.

**Conflicts of Interest:** The authors declare no conflict of interest.

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
