# Peer review of "Optimization of Plant Growth Regulators for In Vitro Mass Propagation of a Disease-Free ‘Shine Muscat’ Grapevine Cultivar"

_cimb, doi:10.3390/cimb45100487_

Round 1

Reviewer 1 Report

The manuscript by Kim et al. describes an in vitro propagation method for producing disease-free grapevine plants. The authors optimized the plant growth regulators concentrations and supplemented their study with ISSR marker analysis to validate the genetic stability of the regenerated plants. Although in vitro propagation methods have already been well established in various plant species, the current study was appropriately designed and the results/conclusions are clearly presented. I have the following suggestions to the authors in order to improve the quality of the manuscript: 

-  I believe that Fig.2, as depicting the summary of the experiment, should be placed at the beginning of the manuscript 

-  I suggest to the authors to include a few characteristic photos of their tissue cultures (e.g. at the lowest and the highest concentration of cytokinin or auxin) at the various stages (e.g. shoot and root formation)

- line 119, "...ranging from 1.0 to 16.0 mΜ and compared with...) -  I think that is μΜ instead of mM

- line 217, "the importance of cytokinins in the initiation of culture and shoot induction" was not demonstrated by this study, it has been already known for decades - please find another sentence to begin this paragraph

- lines 273 - 277 "Another study reported that (...) healthy green leaves" - please include the citation of this study, both in the text and in the reference list

- line 346, "Out of the 10 ISSR primers tested, 11 successfully amplified genomic DNA", I think that this sentence is not correct

- lines 248, 302 and 308 should not be in red

Author Response

Reviewer 1

Comments and Suggestions for Authors

The manuscript by Kim et al. describes an in vitro propagation method for producing disease-free grapevine plants. The authors optimized the plant growth regulator concentrations and supplemented their study with ISSR marker analysis to validate the genetic stability of the regenerated plants. Although in vitro propagation methods have already been well established in various plant species, the current study was appropriately designed and the results/conclusions are clearly presented. I have the following suggestions to the authors in order to improve the quality of the manuscript:

Comment 1: - I believe that Fig.2, as depicting the summary of the experiment, should be placed at the beginning of the manuscript

Response: We would like to express our gratitude for your thoughtful review of our manuscript. We have carefully considered your suggestion regarding the placement of Figure 2.  We changed its placement in the revised manuscript and we have submitted an additional graphical abstract that shows the summary of the experiment.

Comment 2: - I suggest to the authors to include a few characteristic photos of their tissue cultures (e.g. at the lowest and the highest concentration of cytokinin or auxin) at the various stages (e.g. shoot and root formation)

Response: Thank you for your valuable feedback on our manuscript. We appreciate your input and have carefully considered your comments and included Figure 2 in the manuscript.

Comment 3: - line 119, "...ranging from 1.0 to 16.0 mΜ and compared with...) -  I think that is μΜ instead of mM

Response: Thank you for your valuable feedback on our manuscript. We appreciate your careful review and have addressed your comment regarding the unit of concentration in line 119.

Comment 4: - line 217, "the importance of cytokinins in the initiation of culture and shoot induction" was not demonstrated by this study, it has been already known for decades - please find another sentence to begin this paragraph

Response: Thank you for your valuable feedback and for pointing out the need for revision in our manuscript. We appreciate your input and have carefully considered your comments.

Comment 5: - lines 273 - 277 "Another study reported that (...) healthy green leaves" - please include the citation of this study, both in the text and in the reference list

Response: Thank you very much for your valuable feedback.  We have incorporated the citation of this study in both the text and the reference list as requested.

“ Ouyang, Y., Chen, Y., Lü, J., Teixeira da Silva, J.A., Zhang, X. and Ma, G., 2016. Somatic embryogenesis and enhanced shoot organogenesis in Metabriggsia ovalifolia WT Wang. Scientific Reports, 6(1), p.24662.”

Comment 6: - line 346, "Out of the 10 ISSR primers tested, 11 successfully amplified genomic DNA", I think that this sentence is not correct

Response: Thank you for your valuable feedback on our manuscript. We appreciate your attention to detail. We have corrected it accordingly.

Comment 7: - lines 248, 302, and 308 should not be in red

Response: Thank you for your feedback on our manuscript. We have carefully reviewed your comments and have addressed the concerns raised regarding lines 248, 302, and 308 in the revised manuscript.

Reviewer 2 Report

The authors explored the in vitro propagation method for producing disease-free grapes based on the grapevine variety ‘Shine Muscat’ in Korea with the goal to identify effective plant growth regulators to facilitate large-scale cultivation. The effective use of 2.0 μM 6-benzyladenine (BA) and indole-3-butyric acid (IBA) in this effort was further verified by the inter simple sequence repeat analysis confirming the genetic stability of in vitro cultivated seedlings and validating the suitability of the developed propagation method for generating disease-free 'Shine Muscat' grapes.

I have no major technical concern but a few minor suggestions as listed below for the authors to consider if a revision is requested by the editor.

Title: I would suggest that the authors should indicate that the “Shine Muscat” is a variety of grapevine.

Abstract: Line 20, Does “Korea” indicate “South Korea” or “North Korea”?

Introduction: I believe that the authors have provided sufficient background

Materials and Methods: I believe that the authors explained the methodologies well. However, I would like to suggest that the authors provide some pictures to explicitly show the plant materials described in M&M.

Results and Discussion: I believe that the authors presented the results using appropriate tables and figure Again, I would like to suggest that the authors provide some pictures to explicitly show the plant materials described in this section.

Conclusions: I would highly recommend that the authors move Figure 2 and relevant descriptions to the end of Section Results and Discussion simply because it is not appropriate to present Figure 2 in the Section of Conclusion.

Additionally, for unknown reason, some sentences at several places are highlighted in red.

The English is acceptable.

Author Response

Reviewer 2

The authors explored the in vitro propagation method for producing disease-free grapes based on the grapevine variety ‘Shine Muscat’ in Korea with the goal of identifying effective plant growth regulators to facilitate large-scale cultivation. The effective use of 2.0 μM 6-benzyl adenine (BA) and indole-3-butyric acid (IBA) in this effort was further verified by the inter simple sequence repeat analysis confirming the genetic stability of in vitro cultivated seedlings and validating the suitability of the developed propagation method for generating disease-free 'Shine Muscat' grapes.

I have no major technical concern but a few minor suggestions as listed below for the authors to consider if a revision is requested by the editor.

Comment 1: Title: I would suggest that the authors should indicate that the “Shine Muscat” is a variety of grapevine.

Response: Thank you for your valuable feedback on our manuscript titled. We appreciate your time and effort in reviewing our work, and we have carefully considered your suggestions. We incorporated the term “cultivar” as “Shine Muscat” is a cultivated variety.

Comment 2: Abstract: Line 20, Does “Korea” indicate “South Korea” or “North Korea”?

Response: Thank you for your valuable feedback on our manuscript. We appreciate your attention to detail and your effort to ensure clarity in our work. We have corrected it accordingly to “South Korea”.

Comment 3: Introduction: I believe that the authors have provided sufficient background

Response:: Thank you very much

Comment 4: Materials and Methods: I believe that the authors explained the methodologies well. However, I would like to suggest that the authors provide some pictures to explicitly show the plant materials described in M&M.

Response: Thank you for your valuable feedback on our manuscript. We have incorporated your suggestion accordingly in figure 1 and 2.

Comment 5:  Results and Discussion: I believe that the authors presented the results using appropriate tables and figure Again, I would like to suggest that the authors provide some pictures to explicitly show the plant materials described in this section.

Response: Thank you for your valuable feedback on our manuscript regarding the presentation of results and discussion. We appreciate your thorough review and constructive suggestions. We have incorporated your suggestion accordingly in figure 1 and 2.

Comment 6: Conclusions: I would highly recommend that the authors move Figure 2 and relevant descriptions to the end of Section Results and Discussion simply because it is not appropriate to present Figure 2 in the Section of Conclusion.

Response: Thank you for your valuable feedback on our manuscript. We appreciate your thoughtful comments, and we have fully accepted your suggestion and we changed the placement of Figure 2 in the revised manuscript and we have submitted an additional graphical abstract that shows the summary of the experiment.

Comment 7: Additionally, for unknown reasons, some sentences at several places are highlighted in red.

Response: We would like to express our sincere gratitude for your thorough review of our manuscript. We corrected it accordingly.